Submission

# Precise quantum-geometric electronic properties from first principles

**José Luís Martins**[1,2][*]**, Carlos L. Reis**[2] **and Ivo Souza**[3,4][†]

**1** Departamento de Física, Instituto Superior Técnico, Universidade de Lisboa, 1049-001 Lisboa, Portugal
**2** INESC MN, 1000-029 Lisboa, Portugal
**3** Centro de Física de Materiales, Universidad del País Vasco, 20018 San Sebastían, Spain
**4** Ikerbasque Foundation, 48013 Bilbao, Spain

[*] jlmartins@inesc-mn.pt , [†] ivo.souza@ehu.es

## Abstract

The calculation of quantum-geometric properties of Bloch electrons – Berry curvature, quantum metric, orbital magnetic moment and effective mass – was implemented in a pseudopotential plane-wave code. The starting point was the first derivative of the periodic part of the wavefunction $\psi(k)$ with respect to the wavevector k. This was evaluated with perturbation theory by solving a Sternheimer equation, with special care taken to deal with degenerate levels. Comparison of effective masses obtained from perturbation theory for silicon and gallium arsenide with carefully-converged numerical second derivatives of band energies confirms the high precision of the method. Calculations of quantum-geometric quantities for gapped graphene were performed by adding a bespoke symmetry-breaking potential to first-principles graphene. As the two bands near the opened gap are reasonably isolated, the results could be compared with those obtained from an analytical two-band model, allowing to assess the strengths and limitations of such widely-used models. The final application was trigonal tellurium, where some quantum-geometric quantities flip sign with chirality.

# 1   Introduction

Exploring the quantum-geometric and topological properties of electrons in crystals is a very active research topic [1–3]. Band effective masses, which have been a staple of semiconductor physics since the early days of solid-state physics, are now recognized as one of the basic quantum-geometric properties of materials, along with the Berry curvature, quantum metric, and intrinsic orbital moment. A characteristic of such quantities is that they can vary by several orders of magnitude across the Brillouin zone (BZ), with large absolute values usually associated with small regions near band degeneracies or quasi-degeneracies. Therefore, a very fine sampling of the BZ may be required to find those regions, or to calculate properties that are given by BZ integrals of expressions involving quantum-geometric quantities [4]. Interpolation methods, in particular Wannier interpolation, have been extensively used to calculate such properties [5–7], although the evaluation of the quantum metric was implemented only recently [8]. In some cases – for example, the band edges of semiconductors – the region of interest in the BZ may be known, allowing the calculation to be done without recourse to an interpolation scheme.

Effective masses, the inverse of the second derivatives of band energies with respect to wavevector, can be calculated by finite differences [9]. However, in many cases the spin-orbit interaction introduces strong localized perturbations in the band dispersions. Under those circumstances it becomes nontrivial to find good parameters for the numerical differentiation, and to disentangle real from avoided band crossings [10]. Adjusting empirical parameters of $\mathbf{k} \cdot \mathbf{p}$-based models is another possibility, although it requires the prior step of setting up an effective Hamiltonian [11]. Recently, degenerate perturbation theory was used to calculate precise effective masses from first principles with pseudopotential and linearized augmented-plane-wave (LAPW) codes [9, 12], avoiding the difficulties with numerical differentiation or model fitting.

Here, the effective masses are viewed as just one of several related quantum-geometric quantities that can be calculated from the first wavevector derivatives of the cell-periodic Bloch wavefunctions. Other quantities that are obtainable in this way include the Berry curvature, the quantum metric, and the orbital magnetic moment [1, 2], and finite-difference schemes for computing them are available [13, 14]. While the band effective masses and the quantum metric are generically nonzero, the Berry curvature and the orbital magnetic moment vanish identically if both spatial-inversion and time-reversal symmetry are present [1]. In this work, the evaluation of all these quantities is implemented in a pseudopotential plane-wave code using perturbation theory. The code is open source and available for download [15].

The paper is organized as follows. In Sec. 2 the context, method and implementation are presented, paying special attention to the treatment of band degeneracies. To illustrate the

method, the precise effective masses in centrosymmetric silicon and noncentrosymmetric gallium arsenide are presented in Sec. 3; the convergence of the finite-differences and $\mathbf{k} \cdot \mathbf{p}$ methods is also discussed, using our perturbative implementation as a reference. Section 4 deals with an artificial system obtained by breaking the inversion symmetry of graphene, thereby opening a gap at the Dirac points. This allows for a detailed comparison of the calculated *ab initio* quantum-geometric quantities with analytical results from a two-band effective model. The final example, in Sec. 5, is trigonal tellurium, a chiral semiconducting crystal with striking electronic responses driven by quantum geometry at the band edge [16–22]. The article concludes in Sec. 6 with a summary.

## 2 Method

### 2.1 Periodic potential

Once the Kohn-Sham equations [23] of density-functional theory (DFT) [24] are solved and an effective potential $v_{\mathrm{eff}}(\mathbf{r})$ is determined within some approximation, one obtains the Schrödinger equation for that potential,

$$H\psi_{dn}(\mathbf{r}) = \left(-\frac{1}{2}\nabla^2 + v_{\mathrm{eff}}(\mathbf{r})\right)\psi_{dn}(\mathbf{r}) = E_n\psi_{dn}(\mathbf{r}), \tag{1}$$

with one-electron energy levels $E_n$ and wavefunctions $\psi_{dn}(\mathbf{r})$. The energies are assumed to be in strictly increasing order, $E_1 < E_2 < E_3 < \ldots$, and each energy level has degeneracy $D_n$, with $d = 1, 2, \ldots, D_n$ an aditional wavefunction index. Wavefunctions within a degenerate subspace are chosen to be orthonormal: $\langle\psi_{dn}|\psi_{d'm}\rangle = \delta_{dd'}\delta_{nm}$. Here and throughout the paper, we use atomic units ($\hbar = m_e = |e| = 4\pi\epsilon_0 = 1$) to simplify the notation, unless stated otherwise.

For a perfect crystal the potential $v_{\mathrm{eff}}$ is periodic, and one can use Bloch's theorem to choose solutions of the Schrödinger equation that transform according to the representations of the translation symmetry group, labeled by the wavevector $\mathbf{k}$,

$$H|\psi_{dn\mathbf{k}}\rangle = E_{n\mathbf{k}}|\psi_{dn\mathbf{k}}\rangle. \tag{2}$$

In this section, the notation for subscripts will follow the rule of going from the more specific to the more general. For example, in the above equation, the subscripts indicate the function $d$ in the eventually degenerate energy level $n$ of wavevector $\mathbf{k}$. For any given reciprocal-lattice vector $\mathbf{G}$, the representations $\mathbf{k}$ and $\mathbf{k} + \mathbf{G}$ are the same and the energy bands are periodic in reciprocal space,

$$E_{n\mathbf{k}} = E_{n\mathbf{k}+\mathbf{G}}. \tag{3}$$

The Bloch wavefunction can be expressed as the product of a periodic function $u_{dn\mathbf{k}}(\mathbf{r})$ and a spatial phase,

$$\psi_{dn\mathbf{k}}(\mathbf{r}) = e^{i\mathbf{k}\cdot\mathbf{r}}u_{dn\mathbf{k}}(\mathbf{r}). \tag{4}$$

If $\mathbf{R}$ is a translation vector of the lattice, then $u_{dn\mathbf{k}}(\mathbf{r}) = u_{dn\mathbf{k}}(\mathbf{r} + \mathbf{R})$. The cell-periodic parts of the Bloch wavefunctions are also orthonormal, $\langle u_{dn\mathbf{k}}|u_{md'\mathbf{k}}\rangle = \delta_{nm}\delta_{dd'}$, and they satisfy the following eigenvalue equation,

$$H_{\mathbf{k}}|u_{dn\mathbf{k}}\rangle = \left[-\frac{1}{2}\left(\nabla^2 + 2i\mathbf{k}\cdot\boldsymbol{\nabla} - k^2\right) + v_{\mathrm{eff}}\right]|u_{dn\mathbf{k}}\rangle = E_{n\mathbf{k}}|u_{dn\mathbf{k}}\rangle. \tag{5}$$

In the following discussion, we may omit the "periodic part" qualifier when referring to $u_{dn\mathbf{k}}(\mathbf{r})$ if the meaning is clear from the context.

89    In the present work we make the pseudopotential approximation, which allows the use of
90 a plane-wave basis set at the cost of introducing a nonlocal pseudopotential [25]. Following
91 Kleinman and Bylander (KB) [26], the nonlocal part of the pseudopotential is represented by
92 (**k**-dependent) "KB projectors" $|p_{j\mathbf{k}}\rangle$, and the final form of the eigenvalue equation to be solved
93 is

$$H_{\mathbf{k}}|u_{dn\mathbf{k}}\rangle = \left[ -\frac{1}{2}\big(\nabla^2 + 2i\mathbf{k}\cdot\boldsymbol{\nabla} - k^2\big) + v_{\text{eff}} + \sum_j |p_{j\mathbf{k}}\rangle w_j \langle p_{j\mathbf{k}}| \right] |u_{dn\mathbf{k}}\rangle = E_{n\mathbf{k}}|u_{dn\mathbf{k}}\rangle. \qquad (6)$$

94    The weights $w_j$ can be positive or negative, and may be assumed to be just $\pm 1$ if the KB
95 projectors are not normalized.
96    The Bloch eigenfunctions are not unique: if $\{|u_{dn\mathbf{k}}\rangle\}$ is a set of solutions of Eq. (6) for a
97 given energy level $E_{n\mathbf{k}}$, then the states

$$|\tilde{u}_{dn\mathbf{k}}\rangle = \sum_{d'=1}^{D_{n\mathbf{k}}} |u_{d'n\mathbf{k}}\rangle U_{d'd}(n,\mathbf{k}), \qquad (7)$$

98 with $U_{d'd}(n,\mathbf{k})$ an arbitrary unitary matrix, are also solutions of Eq. (6) satisfying the orthonor-
99 mality constraints. This gauge arbitrariness has to be acknowledged when considering the **k**
100 dependence of the wavefunctions.

## 2.2 Perturbation theory

102 Approximations for energies and wavefunctions in the neighborhood of a **k** point can be ob-
103 tained from Eq. (6) using perturbation theory. The main ingredients are the derivatives of the
104 Hamiltonian $H_{\mathbf{k}}$ with respect to the Cartesian components $k_\alpha$ of the **k** vector. We will use Greek
105 letters to indicate Cartesian directions, and place Cartesian indices as subscripts or superscripts
106 in accordance with Einstein convention. The first derivative of the Hamiltonian reads

$$H_{\mathbf{k}}^\alpha \equiv \partial_\alpha H_{\mathbf{k}} = -i\frac{\partial}{\partial r^\alpha} + k_\alpha + \Big( \sum_j |p_{j\mathbf{k}}^\alpha\rangle w_j \langle p_{j\mathbf{k}}| + \sum_j |p_{j\mathbf{k}}\rangle w_j \langle p_{j\mathbf{k}}^\alpha| \Big), \qquad (8)$$

107 where $\partial_\alpha = \partial/\partial k_\alpha$ and $|p_{j\mathbf{k}}^\alpha\rangle = \partial_\alpha |p_{j\mathbf{k}}\rangle$. The second derivative is[1]

$$\begin{aligned}
H_{\mathbf{k}}^{\alpha\beta} \equiv \partial_\alpha\partial_\beta H_{\mathbf{k}} = \delta^{\alpha\beta} + \Big( &\sum_j |p_{j\mathbf{k}}^{\alpha\beta}\rangle w_j \langle p_{j\mathbf{k}}| + \sum_j |p_{j\mathbf{k}}^\alpha\rangle w_j \langle p_{j\mathbf{k}}^\beta| \\
+ &\sum_j |p_{j\mathbf{k}}\rangle w_j \langle p_{j\mathbf{k}}^{\alpha\beta}| + \sum_j |p_{j\mathbf{k}}^\beta\rangle w_j \langle p_{j\mathbf{k}}^\alpha| \Big),
\end{aligned} \qquad (9)$$

108 where $|p_{j\mathbf{k}}^{\alpha\beta}\rangle = \partial_\alpha\partial_\beta |p_{j\mathbf{k}}\rangle$.
109    The first-order equation of degenerate perturbation theory is [2, 27]

$$(E_{n\mathbf{k}} - H_{\mathbf{k}})|u_{dn\mathbf{k}}^\alpha\rangle = \big(H_{\mathbf{k}}^\alpha - E_{dn\mathbf{k}}^\alpha\big)|u_{dn\mathbf{k}}\rangle, \qquad (10)$$

110 with $E_{dn\mathbf{k}}^\alpha$ and $|u_{dn\mathbf{k}}^\alpha\rangle$ the first-order corrections to the energies and wavefunctions, respectively.
111 Multiplying Eq. (10) from the left by $\langle u_{d'n\mathbf{k}}|$ leads to the condition

$$0 = \langle u_{d'n\mathbf{k}}|H_{\mathbf{k}}^\alpha|u_{dn\mathbf{k}}\rangle - E_{dn\mathbf{k}}^\alpha \delta_{dd'}. \qquad (11)$$

---

[1]The derivatives of the pseudopotential projectors are calculated in the code by just applying the chain rule, with some care on how intermediate quantities are defined to ensure easy parallelization. The code uses lattice coordinates, and therefore a metric matrix appears in the expressions; for example, the Kronecker delta in Eq. (9) becomes a metric tensor. Any apparent inconsistencies with the Einstein notation in equations in this paper are indicative of the presence of the metric in the code.

The first-order corrections to the band energies are therefore obtained by diagonalizing the matrix of the perturbation in the degenerate subspace of dimension $D_{n\mathbf{k}}$.

The operator $E_{n\mathbf{k}} - H_{\mathbf{k}}$ has a null space, and the components of $|u_{dn\mathbf{k}}^{\alpha}\rangle$ on that subspace cannot be determined at this level of perturbation theory [27]. However, those components are not needed to calculate the quantum-geometric quantities to be introduced in Sec. 2.4. For that purpose, it is convenient to define for each energy level $E_{n\mathbf{k}}$ the projector onto its eigenspace, which is the null space of $E_{n\mathbf{k}} - H_{\mathbf{k}}$, as well as the projector onto the complement space,

$$P_{n\mathbf{k}} = \sum_{d=1}^{D_{n\mathbf{k}}} |u_{dn\mathbf{k}}\rangle\langle u_{dn\mathbf{k}}|, \qquad Q_{n\mathbf{k}} = \mathbb{1} - P_{n\mathbf{k}}. \tag{12}$$

With these projectors in hand, it is now possible to replace Eq. (10) for $|u_{dn\mathbf{k}}^{\alpha}\rangle$ with a Sternheimer equation for $Q_{n\mathbf{k}}|u_{dn\mathbf{k}}^{\alpha}\rangle$ [2],

$$(E_n - H_{\mathbf{k}})Q_{n\mathbf{k}}|u_{dn\mathbf{k}}^{\alpha}\rangle = Q_{n\mathbf{k}}H_{\mathbf{k}}^{\alpha}|u_{dn\mathbf{k}}\rangle. \tag{13}$$

Thanks to the disappearance of the highly nonlinear first-order energy term $E_{dn\mathbf{k}}^{\alpha}$ from the right-hand side, this is now a linear equation; therefore, any linear combination of the right-hand side will have as a solution the same linear combination of wavefunction perturbations. This means that once solutions have been found for three linearly independent directions, the solution for any direction can be recovered.

With the wavefunctions expanded in a basis set, Eq. (13) is just a linear system, which can in principle be handled by any linear-algebra package. However, when using plane waves the basis sets tend to be very large. Nevertheless, the calculation of $H_{\mathbf{k}}|u\rangle$ for an arbitrary wavefunction $|u\rangle$ can still be done very efficiently [28, 29]. This suggests an iterative solution method for Eq. (13), requiring an appropriate initial guess and a stable iterative procedure. Since that equation has the known closed-form solution [2]

$$Q_{n\mathbf{k}}|u_{dn\mathbf{k}}^{\alpha}\rangle = \sum_{m \neq n} \sum_{d'=1}^{D_{m\mathbf{k}}} \frac{|u_{md'\mathbf{k}}\rangle\langle u_{md'\mathbf{k}}|}{E_{m\mathbf{k}} - E_{n\mathbf{k}}} H_{\mathbf{k}}^{\alpha}|u_{dn\mathbf{k}}\rangle, \tag{14}$$

a truncation of the sum over the energy levels $m$ provides a good starting point for the iterative procedure. A conjugate gradient method that only requires the computation of $H_{\mathbf{k}}|u\rangle$ for an arbitrary $|u\rangle$ will be computationally efficient.

The conjugate gradient method for linear systems is only guaranteed to be stable if the matrix is positive- or negative-definite, which is not the case for Eq. (13) except when $n$ is the lowest energy level. Fortunately, it is easy to recast the equation in a form that is stable, as follows [9]: Define, for fixed $n, d, \alpha$ and a cutoff $N \geq n$, a decomposition of the desired wavefunction derivatives into an inner space and an outer space,

$$Q_{n\mathbf{k}}|u_{dn\mathbf{k}}^{\alpha}\rangle = |u_{dn\mathbf{k}}^{\alpha}\rangle_{\text{in}} + |u_{dn\mathbf{k}}^{\alpha}\rangle_{\text{out}}. \tag{15}$$

In the inner space, we use a truncation of the closed-form solution of Eq. (14),

$$|u_{dn\mathbf{k}}^{\alpha}\rangle_{\text{in}} = \sum_{\substack{m=1 \\ m \neq n}}^{N} \sum_{d=1}^{d_m} \frac{|u_{md\mathbf{k}}\rangle\langle u_{md\mathbf{k}}|}{E_{n\mathbf{k}} - E_{m\mathbf{k}}} H_{\mathbf{k}}^{\alpha}|u_{dn\mathbf{k}}\rangle. \tag{16}$$

Defining the $\mathbf{k}$-dependent projectors

$$P_{\text{in}} = \sum_{n=1}^{N} \sum_{d=1}^{D_{n\mathbf{k}}} |u_{dn\mathbf{k}}\rangle\langle u_{dn\mathbf{k}}|, \qquad Q_{\text{in}} = \mathbb{1} - P_{\text{in}} = P_{\text{out}}, \tag{17}$$

one obtains a Sternheimer equation for $|u_{dn\mathbf{k}}^{\alpha}\rangle_{\text{out}} = Q_{\text{in}}|u_{dn\mathbf{k}}^{\alpha}\rangle$,

$$(E_{n\mathbf{k}} - H_{\mathbf{k}})Q_{\text{in}}|u_{dn\mathbf{k}}^{\alpha}\rangle = Q_{\text{in}}H_{\mathbf{k}}^{\alpha}|u_{dn\mathbf{k}}\rangle. \tag{18}$$

Since for $n \leq N$ the operator on the left-hand side is represented by a negative-definite matrix, the above linear equation is stable with a conjugate gradient solver. In practice, $N$ is chosen as the highest band for which the quantum-geometric quantities of interest are to be calculated.

## 2.3 Band velocities

For each energy level $E_{n\mathbf{k}}$, one can build at first order in perturbation theory the following "vector of matrices," or tensor with three indices, which already appeared in Eq. (11),

$$\epsilon_{dd'}^{\alpha}(n, \mathbf{k}) = \langle u_{dn\mathbf{k}}|H_{\mathbf{k}}^{\alpha}|u_{d'n\mathbf{k}}\rangle, \qquad d, d' = 1, \ldots, D_{n\mathbf{k}}. \tag{19}$$

In the nondegenerate case ($D_{n\mathbf{k}} = 1$), this immediately gives the first derivative of the energy with respect to $k_{\alpha}$, i.e., the band velocity

$$E_{n\mathbf{k}}^{\alpha} = \partial_{\alpha}E_{n\mathbf{k}} = \langle u_{n\mathbf{k}}|H_{\mathbf{k}}^{\alpha}|u_{n\mathbf{k}}\rangle, \tag{20}$$

where the redundant $d$ index was omitted.

When $D_{n\mathbf{k}} > 1$ for a given energy level $E_{n\mathbf{k}}$, it is necessary to first pick a direction $\hat{\mathbf{q}}$ in $\mathbf{k}$ space (with $|\hat{\mathbf{q}}| = 1$), and then construct the first-order perturbation matrix

$$A_{dd'}^{(1)}(\hat{\mathbf{q}}, n, \mathbf{k}) = \sum_{\alpha} \hat{q}_{\alpha}\epsilon_{dd'}^{\alpha}(n, \mathbf{k}), \tag{21}$$

whose eigenvalues $E_{jn\mathbf{k}}^{(1)}(\hat{\mathbf{q}})$ are the band velocities along that particular direction. (Equation (21) is the matrix representation of the first-order Hamiltonian $H_{\mathbf{k}}^{\hat{\mathbf{q}}} = \sum_{\alpha} \hat{q}_{\alpha}H_{\mathbf{k}}^{\alpha}$ within the degenerate subspace.)

## 2.4 Quantum-geometric quantities

Once we have $H_{\mathbf{k}}$, $H_{\mathbf{k}}^{\alpha}$, $H_{\mathbf{k}}^{\alpha\beta}$, and $Q_{n\mathbf{k}}|u_{dn\mathbf{k}}^{\alpha}\rangle$, we can calculate several quantum-geometric quantities, including effective masses, with just algebraic operations. Given a group of $D_{n\mathbf{k}}$ degenerate bands at $\mathbf{k}$, one can associate with it a quantum-geometric tensor defined as [2, 3, 13]

$$T_{dd'}^{\alpha\beta}(n, \mathbf{k}) = \langle u_{dn\mathbf{k}}^{\alpha}|Q_{n\mathbf{k}}|u_{d'n\mathbf{k}}^{\beta}\rangle, \tag{22}$$

which transforms covariantly under the gauge transformation of Eq. (7). This tensor is Hermitian with respect to the interchange of pairs of indices $(d, \alpha) \leftrightarrow (d', \beta)$. It can be split into symmetric and antisymmetric parts in the Cartesian indices, or equivalently, into Hermitian and anti-Hermitian parts in the degeneracy indices. The non-Abelian Berry curvature is

$$\Omega_{dd'}^{\alpha\beta}(n, \mathbf{k}) = i\,T_{dd'}^{\alpha\beta}(n, \mathbf{k}) - i\,T_{dd'}^{\beta\alpha}(n, \mathbf{k}) = i\,T_{dd'}^{\alpha\beta}(n, \mathbf{k}) - i\left(T_{d'd}^{\alpha\beta}(n, \mathbf{k})\right)^{*}, \tag{23}$$

and the non-Abelian quantum metric is

$$g_{dd'}^{\alpha\beta}(n, \mathbf{k}) = \frac{1}{2}T_{dd'}^{\alpha\beta}(n, \mathbf{k}) + \frac{1}{2}T_{dd'}^{\beta\alpha}(n, \mathbf{k}) = \frac{1}{2}T_{dd'}^{\alpha\beta}(n, \mathbf{k}) + \frac{1}{2}\left(T_{d'd}^{\alpha\beta}(n, \mathbf{k})\right)^{*}. \tag{24}$$

With these definitions we have

$$T_{dd'}^{\alpha\beta}(n, \mathbf{k}) = g_{dd'}^{\alpha\beta}(n, \mathbf{k}) - i\frac{1}{2}\Omega_{dd'}^{\alpha\beta}(n, \mathbf{k}). \tag{25}$$

For a nondegenerate level, the quantum metric and the Berry curvature become real-symmetric and real-antisymmetric Cartesian tensors, respectively [30,31],

$$g^{\alpha\beta}(n,\mathbf{k}) = \text{Re}\langle u_{n\mathbf{k}}^{\alpha}|u_{n\mathbf{k}}^{\beta}\rangle - \langle u_{n\mathbf{k}}^{\alpha}|u_{n\mathbf{k}}\rangle\langle u_{n\mathbf{k}}|u_{n\mathbf{k}}^{\beta}\rangle. \tag{26}$$

and

$$\Omega^{\alpha\beta}(n,\mathbf{k}) = -2\text{Im}\langle u_{n\mathbf{k}}^{\alpha}|u_{n\mathbf{k}}^{\beta}\rangle. \tag{27}$$

Moreover, the latter can be transformed into a pseudovector as $\Omega^{\gamma}(n,\mathbf{k}) = \varepsilon_{\alpha\beta\gamma}\Omega^{\alpha\beta}(n,\mathbf{k})$.

Another quantity of interest is the "mass-moment tensor"

$$\Gamma_{dd'}^{\alpha\beta}(n,\mathbf{k}) = \langle u_{dn\mathbf{k}}^{\alpha}|Q_{n\mathbf{k}}(H_{\mathbf{k}} - E_{n\mathbf{k}})Q_{n\mathbf{k}}|u_{d'n\mathbf{k}}^{\beta}\rangle, \tag{28}$$

whose trace over orbital indices (spanning the full set of occupied states, not just a degenerate group) appears in a generalized oscillator-strength sum rule with both time-even and time-odd parts [32]. This quantity has the same symmetry properties as $T_{dd'}^{\alpha\beta}$, and can be partitioned along similar lines. The Cartesian antisymmetric (or band anti-Hermitian) part is the intrinsic orbital magnetic moment tensor of the degenerate group of bands [1,33],

$$\mathfrak{m}_{dd'}^{\alpha\beta}(n,\mathbf{k}) = \frac{|e|}{\hbar}\Big(\frac{1}{2i}\Gamma_{dd'}^{\alpha\beta}(n,\mathbf{k}) - \frac{1}{2i}\Gamma_{dd'}^{\beta\alpha}(n,\mathbf{k})\Big), \tag{29}$$

while the Cartesian symmetric (or band Hermitian) part contributes to a generalized effective mass tensor of the band group, which can be conveniently written as follows,

$$\epsilon_{dd'}^{\alpha\beta}(n,\mathbf{k}) = \hbar^2\Big(\frac{1}{m^*}\Big)_{dd'}^{\alpha\beta}(n,\mathbf{k}) = \langle u_{dn\mathbf{k}}|H_{\mathbf{k}}^{\alpha\beta}|u_{d'n\mathbf{k}}\rangle - \big(\Gamma_{dd'}^{\alpha\beta}(n,\mathbf{k}) + \Gamma_{dd'}^{\beta\alpha}(n,\mathbf{k})\big). \tag{30}$$

(Here we restored the constants $\hbar$ and $|e|$ for clarity.) For a nondegenerate level, the orbital magnetic moment and effective mass tensors become [1,33,34]

$$\mathfrak{m}^{\alpha\beta}(n,\mathbf{k}) = -\frac{|e|}{\hbar}\text{Im}\langle u_{n\mathbf{k}}^{\alpha}|H_{\mathbf{k}} - E_{n\mathbf{k}}|u_{n\mathbf{k}}^{\beta}\rangle \tag{31}$$

and

$$E_{n\mathbf{k}}^{\alpha\beta} = \partial_{\alpha}\partial_{\beta}E_{n\mathbf{k}} = \hbar^2\Big(\frac{1}{m^*}\Big)^{\alpha\beta}(n,\mathbf{k}) = \langle u_{n\mathbf{k}}|H_{\mathbf{k}}^{\alpha\beta}|u_{n\mathbf{k}}\rangle - 2\text{Re}\langle u_{n\mathbf{k}}^{\alpha}|H_{\mathbf{k}} - E_{n\mathbf{k}}|u_{n\mathbf{k}}^{\beta}\rangle. \tag{32}$$

Inserting Eq. (14) in Eq. (32), and specializing to a local potential, yields the familiar sum-over-states expression for the inverse effective mass tensor.

## 2.5  Calculation of effective masses in a given direction

In the absence of degeneracies, the calculation of effective masses is straightforward, as in this case the inverse effective mass is just a rank-two Cartesian tensor, Eq. (32). For degenerate levels, the inverse effective mass is a more complicated tensor, Eq. (30), with two indices on the Cartesian coordinates and two additional degeneracy indices. One way to deal with this tensor is to calculate the effective masses for a given direction $\hat{\mathbf{q}}$ and energy level $E_{n\mathbf{k}}$; these are known as the transport equivalent effective masses [9,12,35].

The starting ingredients are the matrix of first derivatives $A_{dd'}^{(1)}(\hat{\mathbf{q}}, n, \mathbf{k})$ defined in Eq. (21), and the corresponding matrix of second derivatives:

$$A_{dd'}^{(2)}(\hat{\mathbf{q}}, n, \mathbf{k}) = \sum_{\alpha,\beta}\hat{q}_{\alpha}\epsilon_{dd'}^{\alpha\beta}(n,\mathbf{k})\hat{q}_{\beta}, \tag{33}$$

194  with $\epsilon_{dd'}^{\alpha\beta}(n, \mathbf{k})$ taken from Eq. (30). To proceed, we must find the unitary transformation
195  $U_{dd'}(\hat{\mathbf{q}}, n, \mathbf{k})$ that diagonalizes $A_{dd'}^{(1)}(\hat{\mathbf{q}}, n, \mathbf{k})$, written concisely as

$$U^\dagger A^{(1)} U = E_{n\mathbf{k}}^{(1)}. \tag{34}$$

196  Here, $E_{n\mathbf{k}}^{(1)}$ is a diagonal matrix with ordered diagonal elements $E_{1n\mathbf{k}}^{(1)}(\hat{\mathbf{q}}) < E_{2n\mathbf{k}}^{(1)}(\hat{\mathbf{q}}) < \ldots < E_{J_{n\mathbf{k}}n\mathbf{k}}^{(1)}(\hat{\mathbf{q}})$.
197  There are $J_{n\mathbf{k}}(\hat{\mathbf{q}})$ distinct eigenvalues of $A^{(1)}(\hat{\mathbf{q}}, n, \mathbf{k})$, each with degeneracy $I_{jn\mathbf{k}}(\hat{\mathbf{q}})$, satisfying
198  $\sum_{j=1}^{J_{n\mathbf{k}}(\hat{\mathbf{q}})} I_{jn\mathbf{k}}(\hat{\mathbf{q}}) = D_{n\mathbf{k}}$. The same transformation must now be applied to the second-order
199  matrix $A_{dd'}^{(2)}(\hat{\mathbf{q}}, n, \mathbf{k})$,

$$U^\dagger A^{(2)} U = \tilde{A}^{(2)}, \tag{35}$$

200  yielding a new second-order matrix $\tilde{A}_{dd'}^{(2)}(\hat{\mathbf{q}}, n, \mathbf{k})$. Finally, for each band velocity $E_{jn\mathbf{k}}^{(1)}$ we extract
201  from the associated rows and columns of $\tilde{A}_{dd'}^{(2)}(\hat{\mathbf{q}}, n, \mathbf{k})$ the submatrix $\tilde{\tilde{A}}_{dd'}^{(2)}(\hat{\mathbf{q}}, j, n, \mathbf{k})$ of dimen-
202  sions $I_{jn\mathbf{k}} \times I_{jn\mathbf{k}}$. The eigenvalues $E_{ijn\mathbf{k}}^{(2)}(\hat{\mathbf{q}})$ of $\tilde{\tilde{A}}_{dd'}^{(2)}(\hat{\mathbf{q}}, j, n, \mathbf{k})$ are the inverse effective masses
203  along $\hat{\mathbf{q}}$,

$$\left. \frac{\partial^2}{\partial \eta^2} E_{n,\mathbf{k}+\eta\hat{\mathbf{q}}}(i, j) \right|_{\eta=0} = E_{ijn\mathbf{k}}^{(2)}(\hat{\mathbf{q}}). \tag{36}$$

204      We note that the procedure outlined above is equivalent to applying the unitary transfor-
205  mation $U$ that diagonalizes $A^{(1)}$ to the original wavefunctions $|u_{jn\mathbf{k}}(\hat{\mathbf{q}})\rangle$ to obtain a new set
206  of wavefunctions $|\tilde{u}_{ijn\mathbf{k}}(\hat{\mathbf{q}})\rangle$, with $i = 1, \ldots, I_{jn\mathbf{k}}(\hat{\mathbf{q}})$ and $j = 1, \ldots, J_{n\mathbf{k}}(\hat{\mathbf{q}})$; the new set is then
207  used to recalculate Eqs. (28) and (30), as they are the relevant wavefunctions for carrying out
208  second-order degenerate perturbation theory [27] in the subspace with first-order eigenvalue
209  corrections $E_{jn\mathbf{k}}^{(1)}(\hat{\mathbf{q}}, n, \mathbf{k})$. However, applying the transformations directly to the operators is
210  computationally more efficient.

211      Effective masses of degenerate bands have been calculated from first principles using per-
212  turbative methods (as opposed to finite differences) with the ABINIT [9] and WIEN2k [12]
213  codes; for the nondegenerate case, there is earlier work [36]. As those calculations, as well
214  as the present work, use perturbation theory to calculate effective masses, and their basis sets
215  are all related to plane waves – linearized augmented-plane-wave (LAPW), projected aug-
216  mented wave (PAW) or just elementary plane waves – there are close similarities between all
217  three implementations. The main distinction between the present approach and the previous
218  work is that we use $Q_{n\mathbf{k}}|u_{dn\mathbf{k}}^\beta\rangle$ given by the Sternheimer equation (13) as our basic ingredi-
219  ent, whereas the previous work is based on explicit sums over states; the connection between
220  the two approaches is Eq. (14). The ABINIT implementation evaluates the effective masses
221  in a plane-wave basis via the expression in Eq. (66) of Ref. [9]. That expression is different
222  from our Eq. (30) with $\Gamma_{dd'}^{\alpha\beta}(n, \mathbf{k})$ defined by Eq. (28), although it is possible to show they
223  are equivalent using Eq. (14). As the ABINIT implementation uses a Sternheimer equation to
224  compute the sum over states, there are also points in common at the algorithmic level. The
225  present approach has the advantage of yielding the Berry curvature, quantum metric and or-
226  bital moment for an almost negligible extra computational cost, presenting the calculation of
227  quantum-geometric quantities in a unified framework.

# 3  Semiconductor effective masses

## 3.1  Silicon

230  As a first example, we study the effective masses of Si, with special emphasis on the top of
231  the valence band at $\Gamma$ ($\mathbf{k} = \mathbf{0}$). Without spin-orbit coupling and ignoring spin, the top of

the valence band consists of three degenerate states with $p$ character around each of the two atoms in the primitive cell. With spin-orbit interaction, spin cannot be ignored, and the six states at $\Gamma$ split into a twice-degenerate level with lower energy (the "split-off hole" band), and four degenerate states which become the true valence-band maximum. Although the spin-orbit splitting at $\Gamma$ is small, it is of the order of the energy associated with room temperature, and therefore it cannot be neglected. As Si has both inversion symmetry and time-reversal symmetry, all bands are doubly-degenerate across the BZ. When moving away from $\Gamma$ there are two (doubly-degenerate) bands with different effective masses, called "light-hole" and "heavy-hole" bands.

The self-consistent potential of Si was calculated in the local-density approximation (LDA) with the Perdew-Wang parametrization [37], using the CPW2000 pseudopotential plane-wave code [15]. The pseudopotential was relativistic Troullier-Martins [38,39] with a core radius of 1.8 atomic units, ground-state configuration, and $s$, $p$, and $d$ channels. The local potential was a smoothed maximum of all channels, and the nonlocal part was converted to the KB form. The self-consistent calculation used a kinetic energy cutoff of 20 Ha for the plane-wave expansion, and a $6 \times 6 \times 6$ uniform grid for BZ integration (28 **k** points once symmetry was taken into account). The lattice constant in the calculations was 5.4015 Å. The lattice constant, energy cutoff, exchange-correlation functional, and BZ sampling were chosen to be the same as those of Ref. [9], so that any differences in results could be attributed to the difference in electronic-structure method: projected augmented wave (PAW) in Ref. [9], versus pseudopotential in the present work. The calculations were well converged: increasing cutoff or BZ sampling had typically a minor effect on the 4th decimal place of the effective masses. Using other LDA parametrizations, or using a different core radius, also only had an effect on the 4th decimal place. Using a generalized gradient approximation (GGA) only slightly changed the third decimal place. However, using the experimental lattice constant changed the effective masses by a few percent. As we will be checking numerical accuracy, in some tables we report a high number of decimal places.

Away from band degeneracies, effective masses are described by the Cartesian tensor of second-order energy derivative, Eq. (32). That real-symmetric tensor is described by its principal axes and inverse masses along those axes. In a cubic system and for a nondegenerate state at $\Gamma$, the tensor would just be a multiple of the identity, with identical effective masses in every direction. In Si, the split-off hole band and the lowest conduction band at $\Gamma$ only have the spin double degeneracy and have isotropic effective masses. However, at the top of the valence band where the bands are four-fold degenerate, the formulation of section 2.5 must be employed. The resulting effective masses are strongly anisotropic, as can be seen in Fig. 1. Similar figures can be found in the literature [35].

Table 1 shows the effective band mass in Si at the $\Gamma$ point for high symmetry directions. States usually described as holes appear with negative masses. States are identified by symmetry, with the added subscript $v$ for valence or $c$ for conduction. The first column contains the results of the present work. They agree within a few percent with the PAW calculations shown in the second column. Effective masses can also be calculated by finite differences from energy bands calculated along a few points in the chosen direction, and we show in that table the results from Lagrange interpolations of order $n = 6$ and 8, with $n + 1$ points centered at $\Gamma$ and spaced by $\delta = 10^{-3}$, $10^{-4}$ or $10^{-5}$ times $2\pi/a$. Instead of an analytical finite-difference formula for the second derivatives, we use a recursive algorithm, that has the advantage of being numerically stable. As an added bonus, by comparing the second derivatives at order $n - 1$ of the recursion with the final value, an indication of the reliability of the estimate is obtained [40]. One can see that for $\delta = 10^{-4}$, the $n = 6$ and $n = 8$ interpolations give the same values for the effective masses as perturbation theory. For the larger $\delta = 10^{-3}$ spacing of the sampling wavevectors, there are some deviations in mass values, but they are still accurate

## Silicon

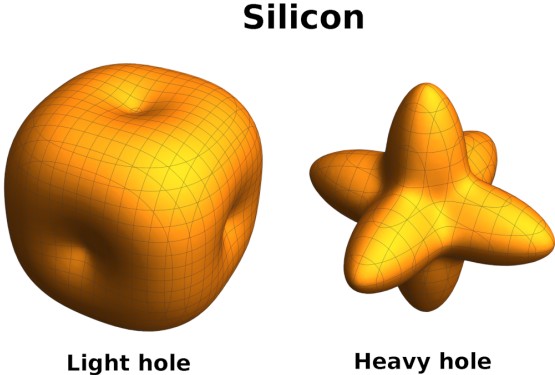

**Light hole**          **Heavy hole**

Figure 1: The dependence on direction of the second derivative of the band energy (inverse effective mass) is shown for the top of the valence band of Si. The distance from the center of each figure to the surface is proportional to the second derivative of the band energy in that direction. The inverse effective masses are clearly anisotropic, and cubic symmetry is respected. The scale of the two figures is not the same.

for the physics of the problem. For the smaller $\delta = 10^{-5}$ spacing, the numerical instability of Lagrange interpolations for closely-spaced points appears, and the results are less reliable. In fact, for that small spacing, the degeneracy of the mass values (they should all be doubly degenerate) is numerically broken, and the values reported in the table are their mean values. The main point of the table is that the implemented method works and is very stable and accurate.

There is an approximation from the early days of electronic structure theory, the $\mathbf{k} \cdot \mathbf{p}$ method, that uses the second-order expansion of the Hamiltonian implicit in Eqs. 1, 8 and 9. (Obviously, the early work did not have the projectors of the non-local pseudopotential.) Given a reference wavevector $\mathbf{k}_0$ and its eigenfunctions $|u_{dn\mathbf{k}_0}\rangle$, one can construct an approximate Hamiltonian matrix for a neighboring wave vector $\mathbf{k}$

$$
\begin{aligned}
H_{dn,d'n'}^{\mathbf{k}\cdot\mathbf{p}} = \langle u_{dn\mathbf{k}_0}|H_{\mathbf{k}_0}|u_{d'n'\mathbf{k}_0}\rangle + \sum_{\alpha}(\mathbf{k}-\mathbf{k}_0)_\alpha \langle u_{dn\mathbf{k}_0}|H_{\mathbf{k}_0}^\alpha|u_{d'n'\mathbf{k}_0}\rangle \\
+ \sum_{\alpha\beta}(\mathbf{k}-\mathbf{k}_0)_\alpha \langle u_{dn\mathbf{k}_0}|H_{\mathbf{k}_0}^{\alpha\beta}|u_{d'n'\mathbf{k}_0}\rangle(\mathbf{k}-\mathbf{k}_0)_\beta,
\end{aligned}
\tag{37}
$$

where indices $n$ and $n'$ span a truncated set of bands. Here the Luttinger-Kohn basis set [41] is used, and the approximation gets its name from the $\mathbf{k}\cdot(-i\boldsymbol{\nabla})$ factor present in the second term of Eq. (5), from which Eq. (37) follows. These are the same ingredients as the perturbation theory described previously; what is missing is the Sternheimer equation that corrects for the band truncation. The advantage of this scheme is that once the matrix elements in the basis set are calculated for $\mathbf{k}_0$, it is trivial to construct $H^{\mathbf{k}\cdot\mathbf{p}}$ for any $\mathbf{k}$ and, since it is a small matrix, diagonalize it. Table 2 compares the effective masses from perturbation theory with those obtained from finite differences with a $\mathbf{k}\cdot\mathbf{p}$ Hamiltonian based on basis sets with 9, 15, 59, 112 and 259 functions (not counting spin). One can see that the 15-band model (30 functions counting spin) already describes with reasonable accuracy (within a few percent) the effective masses. As expected, their values converge to the correct ones with increasing model size.

## 3.2  Gallium arsenide

While the general features of the gallium arsenide band structure are similar to those of silicon, the absence of spatial inversion symmetry means that the bands are not necessarily degenerate,

| | | Perturbation | | Finite differences | | | | |
|---|---|---|---|---|---|---|---|---|
| | | PW-PT | PAW-PT | FD 6, $10^{-3}$ | FD 6, $10^{-4}$ | FD 8, $10^{-4}$ | FD 6, $10^{-5}$ | FD 8, $10^{-5}$ |
| $\Gamma_{6v}$ | | 1.161527 | 1.161530 | 1.161655 | 1.161529 | 1.161528 | 1.161640 | 1.161644 |
| $\Gamma_{7v}$ | | -0.226749 | -0.222588 | -0.226746 | -0.226749 | -0.226749 | -0.226747 | -0.226747 |
| $\Gamma_{8v}$ LH | [100] | -0.191152 | -0.188252 | -0.191151 | -0.191152 | -0.191152 | -0.191162 | -0.191162 |
| | [110] | -0.139122 | -0.136672 | -0.139122 | -0.139122 | -0.139122 | -0.139126 | -0.139280 |
| | [111] | -0.132074 | -0.129739 | -0.132073 | -0.132074 | -0.132074 | -0.132073 | -0.132225 |
| $\Gamma_{8v}$ HH | [100] | -0.260038 | -0.253933 | -0.260032 | -0.260038 | -0.260038 | -0.260034 | -0.260034 |
| | [110] | -0.529353 | -0.517250 | -0.529345 | -0.529353 | -0.529353 | -0.529334 | -0.528980 |
| | [111] | -0.664230 | -0.648382 | -0.664234 | -0.664230 | -0.664230 | -0.664365 | -0.663356 |
| $\Gamma_{6c}$ | | 0.395668 | 0.385387 | 0.395671 | 0.395669 | 0.395669 | 0.395692 | 0.395694 |

Table 1: Effective masses (in atomic units) of Si bands at $\Gamma$. The values calculated in the present work with plane waves and perturbation theory (PW-PT) are compared with the published values obtained using the ABINIT code with projected augmented waves and perturbation theory [9] (PAW-PT), and with our finite-difference (FD) estimates. The finite-difference values are from Lagrange interpolation; the $n$ in the notation "FD n" indicates the order of the polynomial and is followed by the value (in atomic units) of the spacing between the $n+1$ interpolation points.

| | | Perturbation | $\mathbf{k} \cdot \mathbf{p}$ | | | | |
|---|---|---|---|---|---|---|---|
| | | PW-PT | $n = 9$ | $n = 15$ | $n = 59$ | $n = 112$ | $n = 259$ |
| $\Gamma_{6v}$ | | 1.161 | 1.082 | 1.081 | 1.125 | 1.141 | 1.158 |
| $\Gamma_{7v}$ | | -0.227 | -0.249 | -0.232 | -0.230 | -0.228 | -0.227 |
| $\Gamma_{8v}$ LH | [100] | -0.191 | -0.221 | -0.196 | -0.194 | -0.192 | -0.191 |
| | [110] | -0.139 | -0.143 | -0.141 | -0.140 | -0.140 | -0.139 |
| | [111] | -0.132 | -0.135 | -0.134 | -0.133 | -0.133 | -0.132 |
| $\Gamma_{8v}$ HH | [100] | -0.260 | -0.268 | -0.267 | -0.263 | -0.262 | -0.261 |
| | [110] | -0.529 | -0.619 | -0.562 | -0.544 | -0.536 | -0.532 |
| | [111] | -0.664 | -0.742 | -0.717 | -0.687 | -0.675 | -0.668 |
| $\Gamma_{6c}$ | | 0.396 | 0.243 | 0.381 | 0.388 | 0.391 | 0.394 |

Table 2: Effective masses (in atomic units) of Si bands at $\Gamma$. The values calculated with plane-wave perturbation theory (PW-PT) are compared with those obtained by $\mathbf{k} \cdot \mathbf{p}$ models using a basis $n = 9, 15, 59, 112, 259$ orbitals (not counting spin).

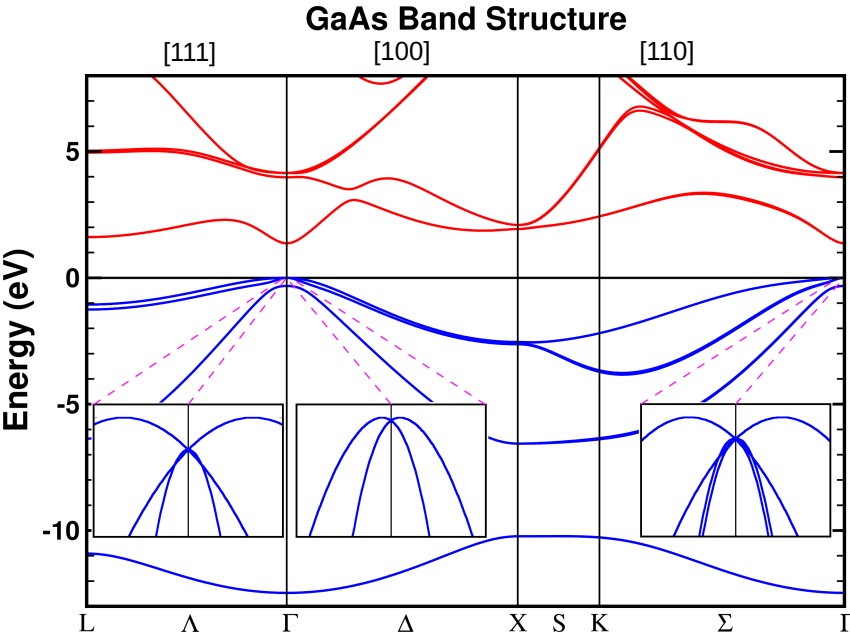

Figure 2: The valence bands (blue) and conduction bands (red) of GaAs are shown for the main symmetry directions of the fcc lattice. The highly magnified behavior of the hole bands near the top of the valence band is shown in the insets, where the energy range is $10^{-5}$ eV and the wavevector range is $1.18 \times 10^{-3}$ atomic units.

and at the top of the valence band at $\Gamma$, the band slope may be non-zero. These effects are easily missed in a band structure plot showing the full energy range of the valence and lower conduction bands, see Fig 2, or even in the scale of room temperature; however, they are clearly seen on the $10^{-5}$ eV energy range of the insets of Fig 2. The perturbation theory at the $\Gamma$ point will give the first and second derivatives of the energy at that point, which is connected with the smallest of the energy scales. This behavior near the top of the valence band of GaAs was described by E. Dresselhaus in a seminal paper [42].

The self-consistent potential of GaAs was calculated with the CPW2000 pseudopotential plane-wave code using a modified Becke-Johnson meta-GGA functional [43] that gives an energy gap close to experiment. Both Ga and As pseudopotentials were relativistic, constructed with the Troullier-Martins recipe [38] from a ground-state configuration and LDA exchange and correlation. The Ga pseudopotential had a core radius of 2.5 atomic units for the $s$ and $p$ channels, and 2.69 atomic units for the $d$ channel. The As pseudopotential had a core radius of 2.1 atomic units for the $s$, $p$ and $d$ channels. In both cases the $d$ channel was used as the local part, and the nonlocal part was converted to the KB form. The self-consistent calculation used a kinetic energy cutoff of 16 Ha for the plane-wave expansion, and a $4 \times 4 \times 4$ uniform grid for BZ integration (10 **k** points once symmetry was taken into account). The lattice constant in the calculations was 5.653 Å.

The calculated effective masses at $\Gamma$ are shown in Table 3. There is an excellent agreement (about six decimal places) between the masses calculated from perturbation theory and those obtained from an interpolation of band energies (with a sixth order polynomial, and points equally spaced by $10^{-5}$ and centered in $\Gamma$). However, in the interpolation, the bands have to be carefully "disentangled" to follow the correct branch after each crossing. Even with the help of the quality estimate of the fit that flags most mistakes [40], that disentanglement is a very tedious and error prone job.

| | | Perturbation | $\mathbf{k}\cdot\mathbf{p}$ | | Finite differences | | |
| --- | --- | --- | --- | --- | --- | --- | --- |
| | | PW-PT | $n=9$ | $n=15$ | FD 3, $10^{-3}$ | FD 3, $10^{-4}$ | FD 3, $10^{-5}$ |
| $\Gamma_{6v}$ | | 1.345679 | 0.837 | 0.838 | 1.345746 | 1.345331 | 1.345676 |
| $\Gamma_{7v}$ | | -0.191713 | -0.222 | -0.195 | -0.191746 | -0.191717 | -0.191714 |
| $\Gamma_{8v}$ LH | [100] | -0.152844 | -0.181 | -0.161 | -0.095675 | -0.152846 | -0.152845 |
| | [110] | -0.088951 | -0.098 | -0.093 | -0.086067 | -0.099610 | -0.088952 |
| | [111] | -0.083548 | -0.091 | -0.088 | -0.083547 | -0.083548 | -0.083548 |
| $\Gamma_{8v}$ HH | [100] | -0.152844 | -0.181 | -0.161 | -0.380019 | -0.152846 | -0.152845 |
| | [110] | -0.542557 | -1.194 | -0.593 | -0.683390 | -0.328346 | -0.542561 |
| | [111] | -0.895971 | — | -0.996 | -0.895868 | -0.895949 | -0.896110 |
| $\Gamma_{6c}$ | | 0.075658 | 0.074 | 0.074 | 0.075646 | 0.075659 | 0.075658 |

Table 3: Effective masses (in atomic units) of the GaAs bands at $\Gamma$. The values calculated with plane-wave perturbation theory (PW-PT) are compared with those obtained by $\mathbf{k}\cdot\mathbf{p}$ with $n=9$ or 15 orbitals (not counting spin), and by finite differences (FD). The notation "FD n" is the same as in Table 1.

A clear sign of the peculiarities in the effective masses is that at $\Gamma$ and in the [100] direction they are identical for the "heavy" and "light" holes. In that direction, the four bands that are degenerate at $\Gamma$ split into two doubly degenerate bands that are mirror images of each other, and therefore have the same second derivatives at $\Gamma$. From the insets of Fig. 2, it is clear that near $\Gamma$ the bands deviate from parabolas. In Fig. 3, the longitudinal masses along [100] are plotted as a function of distance from the $\Gamma$ point. The reciprocal vector is on a scale $x = log(1 + k/k_0)$ that is linear near the origin ($k \ll k_0$) but logarithmic far from it ($k \gg k_0$). The mass of the split-off hole is almost constant throughout the plot range. The heavy and light hole masses are identical at $\Gamma$, deviating linearly at very small $k$ until they settle at an almost constant value.

The GaAs bands near $\Gamma$ illustrate how the degenerate perturbation theory described in the previous section works in practice. In the [111] direction ($\Gamma-L$), diagonalization of the $4 \times 4$ matrix $A^{(1)}(\hat{\mathbf{q}} = [111], n = \text{VBM}, \mathbf{k} = \Gamma)$ of Eq. (21) for the four-fold degenerate valence band maximum (VBM) gives two nonzero and nondegenerate band velocities with opposite signs, and a twice-degenerate vanishing band velocity (see left inset of Fig. 2). For the two nondegenerate band velocities, the masses are obtained from the relevant diagonal elements of $\tilde{A}^{(2)}([111], \text{VBM}, \Gamma)$, which results from applying to $A^{(2)}([111], \text{VBM}, \Gamma)$ the unitary transformation that diagonalizes $A^{(1)}([111], \text{VBM}, \Gamma)$; see Eqs. (33) to (36). These two diagonal elements are identical, implying that the corresponding masses are identical. For the doubly-degenerate level with vanishing band velocity, the masses are obtained by diagonalizing the corresponding $2 \times 2$ submatrix of $\tilde{A}^{(2)}([111], \text{VBM}, \Gamma)$, yielding two identical masses. In the [100] direction ($\Gamma-X$), $A^{(1)}([100], \text{VBM}, \Gamma)$ has two doubly-degenerate nonzero band velocities with opposite signs (see middle inset of Fig. 2). For each sign, the corresponding $2 \times 2$ submatrix of $\tilde{A}^{(2)}([100], \text{VBM}, \Gamma)$ must be diagonalized; the final result is four identical masses as discussed previously in connection with Fig 3. Finally, in the [110] direction ($\Gamma-K$), $A^{(1)}([110], \text{VBM}, \Gamma)$ has four distinct band velocities: two with large absolute values, and two with small absolute values (see right inset of Fig. 2). All the masses are obtained from the corresponding diagonal elements of $\tilde{A}^{(2)}([110], \text{VBM}, \Gamma)$. The fact that at $\Gamma$ the effective mass degeneracies are multiples of two is a consequence of time reversal symmetry.

**GaAs [100] hole effective mass**

Figure 3: The effective masses near the top of the valence bands of GaAs are shown as a function of the deviation from the Γ point. The horizontal scale is "shifted logarithmic" as described in the text, since the system has several energy and wavevector scales. The split-off hole mass is almost constant. The light- and heavy-hole masses are constant over a sizeable wavevector range, but converge to the same value at Γ.

# 4 Gapped graphene

Gapped graphene is a system where the inversion symmetry of the two-dimensional pristine structure is broken by an external potential, typically associated with a substrate on which graphene is grown epitaxially [44,45]. The equivalence between the two atomic sites in the primitive unit cell is broken, and as a result a gap opens in the Dirac cones at the Fermi level. The simplest tight-binding (TB) model [46,47], with just one $p_z$ orbital per site, leads to a two-band low-energy model with just two parameters: the magnitude of the gap $\Delta$ and the Fermi velocity $v_F = \sqrt{3}at/2$ [46], where $a$ is the lattice constant and $t$ the hopping integral of the TB model. Quantum-geometric quantities have simple analytical expressions in two-band models [48,49], and the low-energy model for gapped graphene has been widely used in recent times to investigate such quantities [50].

Adding a bespoke external potential with a gaussian shape to one of the atomic sites in a first-principles calculation for graphene opens such a gap in the Dirac cones. This allows a comparison of the quantum-geometric quantities calculated from first principles with those from the low-energy model. Obviously, one cannot expect the exact same results, but they should be similar, particularly in the limit of a vanishing gap.

The self-consistent potential of broken-symmetry graphene was calculated using a modified Becke-Johnson meta-GGA functional [43] in a supercell configuration. The Tran-Blaha constant [43] was fixed at $c_{TB} = 1.04$, since the original recipe is inadequate for isolated 2D materials. While the LDA approximation is known to underestimate the bonding energy of $\sigma$ electrons with respect to $\pi$ electrons [51], we find that the modified Becke-Johnson meta-GGA functional gives results closer to experiment. A Troullier-Martins [38] pseudopotential was constructed from a local-density ground-state configuration, with a core radius of 1.3 atomic units for the $s$, $p$, and $d$ channels. The local potential was a smoothed maximum of all channels, and the nonlocal part was converted to the Kleynman-Bylander form. The self-consistent calculation used a kinetic energy cutoff of 50 Ha for the plane-wave expansion, and a $6 \times 6 \times 2$ uniform grid for BZ integration. The lattice constant in the calculations was 2.456 Å in the graphene plane, three times that value was used in the perpendicular supercell direction. The added inversion-breaking potential was tweaked to open a gap of 0.28 eV; this is the same value that was used in Ref. [50], and is also very close to the experimental value reported for epitaxial graphene on a SiC substrate [44].

Figure 4 shows the DFT-pseudopotential energy bands as a function of wavevector distance $q$ from K towards Γ, together with the bands of the low-energy model,

$$E_\pm = \pm \frac{\Delta}{2} \sqrt{1 + (q/q_0)^2},    \tag{38}$$

where we defined a characteristic wavevector,

$$q_0 = \frac{\Delta}{\sqrt{3}at} = \frac{\Delta}{2v_F}.    \tag{39}$$

In the paper of Xiao *et. al* [50] the parameters are $\Delta = 0.28$ eV, $a = 2.456$ Å and $t = 2.82$ eV, so that $q_0 \simeq 0.01235$ a.u. With this value of $q_0$, the energy bands of the low-energy model (dashed-black lines in Fig. 4) look fine for $q \lesssim q_0$, which is not surprising as the external potential was chosen to give the same gap value $\Delta$ at K. However, deviations for $q \gg q_0$ are noticeable, meaning that the Fermi velocity of the underlying pseudopotential graphene energy bands (without the added potential) is slightly lower than assumed in Ref. [50]. If we use the effective mass at K from the gapped-graphene DFT calculation as the second constraint in the model (more precisely, the average of the absolute values of the valence and conduction

## Gapped graphene bands

Figure 4: The energy bands of gapped graphene are shown as a function of wavevector distance from (BZ corner) in the direction of the BZ center. The blue squares and red dots are the valence and conduction bands obtained by adding a bespoke potential to the DFT calculation. The dashed-black lines and solid-green lines both pertain to the low-energy model, and are obtained by inserting slightly different sets of parameters into Eq. (38).

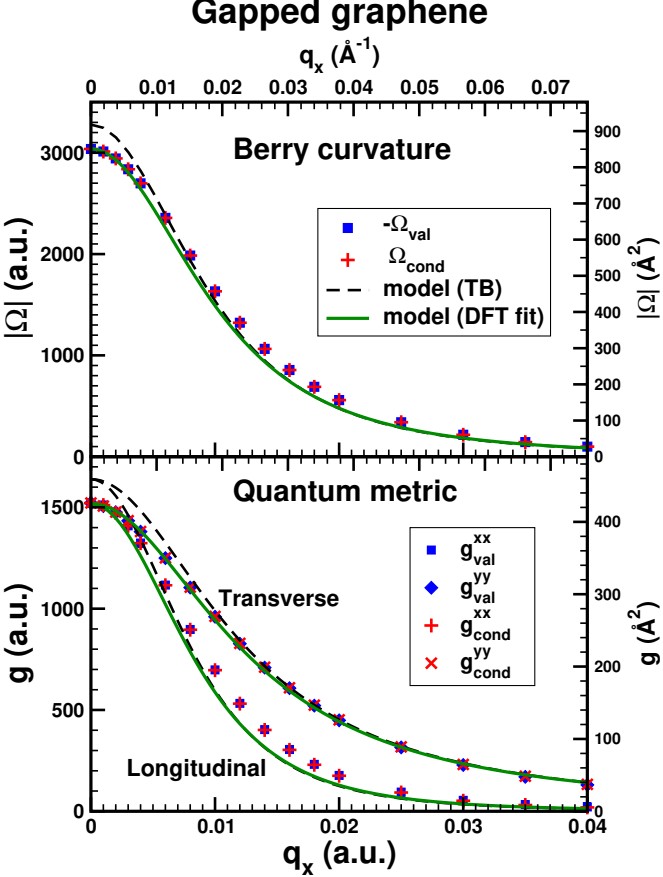

Figure 5: The absolute value of the Berry curvature (top panel) and components of the quantum metric tensor (bottom panel) of gapped graphene are shown as a function of the distance from K towards Γ. The DFT results are shown using blue symbols for the valence band and red symbols for the conduction band. The dashed-black lines and solid-green lines both pertain to the low-energy model and are obtained by inserting slightly different sets of parameters into Eqs. (40) and (41).

effective masses), we obtain a value of $q_0 = 0.01283$ for the characteristic wavevector, resulting in slightly different bands (solid-green lines in Fig. 4) which are closer to the DFT bands for $q \gg q_0$. The agreement is not perfect, and should not be expected to be, but it is very good.

In the top panel of Fig. 5, the Berry curvature $\Omega = \Omega_{xy}$ calculated from DFT is compared with the low-energy model [50],

$$\Omega = \pm \frac{1}{2q_0^2} \frac{1}{[1 + (q/q_0)^2]^{3/2}};  \qquad (40)$$

both quantities are plotted as a function of wavevector distance from K towards Γ. The Berry curvature has opposite signs for the two bands, and we plot the absolute value to emphasize that on the scale of the figure, the differences between the two bands are not visible. The sign of $\Omega$ alternates between the two K points that are not equivalent by translation. It also flips depending on top of which carbon atom in the unit cell the potential is added to, or if that potential is attractive or repulsive. In the low-energy model, $\Delta$ is often considered as a signed quantity, adding a further sign choice. As there is a relationship between the Berry curvature and the effective masses at $q = 0$ in the model [see Eq. (38) and Eq. (40)], it is not surprising that fitting the effective mass at K gives an excellent agreement with the Berry

417   curvature near that point (solid-green line). Overall, the low-energy model can describe quite
418   well the DFT-pseudopotential results in the displayed range.

419      The DFT-pseudopotential quantum metric is compared in the bottom panel of Fig. 5 with
420   that of the low-energy model, given by Eq. (41) below. By symmetry, the off-diagonal compo-
421   nents are null for our choice of axis orientation and plot direction, therefore we only plot the
422   two in-plane diagonal components. In the low-energy model they are given by

$$g^{xx}(q_x, 0) = \frac{1}{4q_0^2} \frac{1}{[1 + (q_x/q_0)^2]^2}$$

$$g^{yy}(q_x, 0) = \frac{1}{4q_0^2} \frac{1}{[1 + (q_x/q_0)^2]^1},$$

(41)

423   with $x$ in the direction from K to Γ. At K $g^{xx} = g^{yy} = |\Omega|/2$ [note the factor-of-two difference
424   in Eqs. (23) and (24)], and the results from the model with the fitted $q_0$ are very close to the
425   DFT values, particularly for $q_x \ll q_0$. As in the case of the Berry curvature, the differences in
426   the DFT values for the valence and conduction bands are so small that they are not noticeable
427   on the scale of the figure. The dependence on $q_x$ is very different for the longitudinal and
428   transverse components of the quantum metric. The model fit of the DFT results is very good for
429   the longitudinal component, but for the transverse component there is a noticeable deviation.
430   As the system has a third dimension, the $zz$ component of the metric has a nonzero value,
431   $g^{zz} = 0.56$ Å$^2$ at $q = 0$, which is very small on the scale of Fig. 5, and depends very weakly on
432   $q_x$.

433      Turning now to the inverse effective masses, in the bottom panel of Fig. 6 the DFT results
434   are compared with those of the low-energy model, which are evaluated as

$$\left(\frac{1}{m^*}\right)^{xx}(q_x, 0) = \frac{1}{m_e} \frac{\Delta}{2q_0^2} \frac{1}{[1 + (q_x/q_0)^2]^{1/2}}$$

$$\left(\frac{1}{m^*}\right)^{yy}(q_x, 0) = \frac{1}{m_e} \frac{\Delta}{2q_0^2} \frac{1}{[1 + (q_x/q_0)^2]^{3/2}},$$

(42)

435   with $m_e$ the electron mass (one in atomic units). The small deviations seen in Fig. 4 between
436   the DFT and model energy bands get magnified in the inverse masses, which are their second
437   derivatives. For $q_x = 0$, there is a small difference in masses between the valence and conduc-
438   tion band that is not apparent on the scale of the figure. As the average effective mass was used
439   for the value of $q_0$ in the DFT fit, the agreement between the model and DFT calculations is
440   not surprising. When moving away from K towards Γ, the longitudinal and transverse inverse
441   masses become different, the longitudinal mass decreasing at a much faster rate.

442      The final quantum-geometric quantity on our list is the orbital magnetic moment, which
443   is plotted in the top panel of Fig. 6. The results for the low-energy model were obtained from
444   the following expression,

$$\mathfrak{m}(q_x, 0) = \pm 2\mu_B \frac{\Delta}{4q_0^2} \frac{1}{[1 + (q_x/q_0)^2]^1}$$

(43)

445   where the Bohr magneton, $\mu_B$, has the value $1/2$ in atomic units. In contrast with previous
446   figures, there is a clear difference between the orbital magnetic moments of the valence and
447   conduction bands at K, although they both have values (in atomic units) close to half the
448   inverse mass, as predicted by the low-energy model. The sign of the orbital magnetic moment
449   is the same for the two bands, but, like the Berry curvature, it depends on which of the two
450   inequivalent K points is considered, or on which atom the symmetry-breaking potential is
451   placed. A consistency check is that the orbital moment must have the same sign as the Berry
452   curvature of the conduction band.

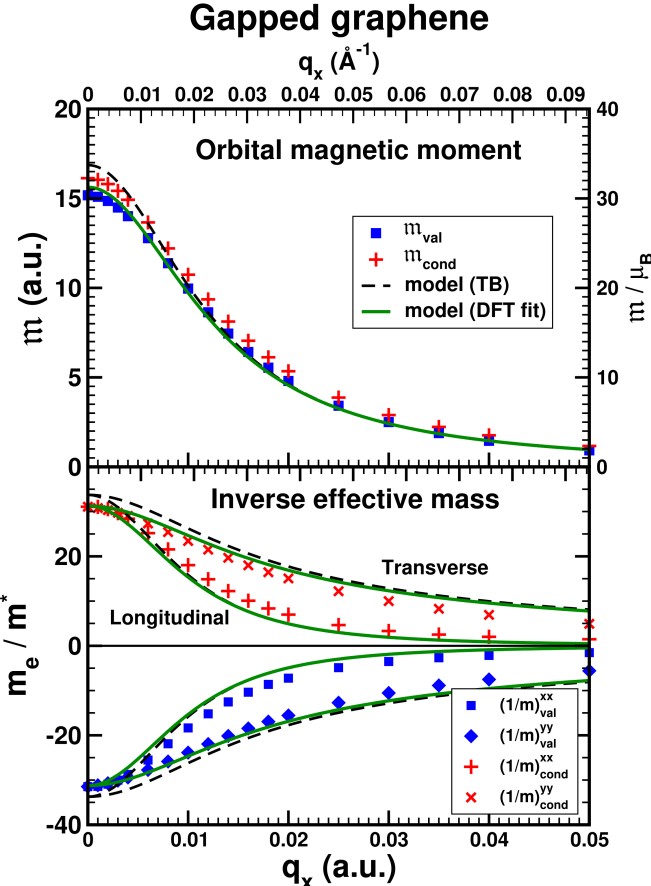

Figure 6: The orbital magnetic moments (top panel) and inverse effective masses (bottom panel) of gapped graphene are shown as a function of the distance from K towards Γ. The longitudinal components of the inverse effective mass tensor are the second derivatives of the energy dispersions in Fig. 4. The DFT results are shown using blue symbols for the valence band and red symbols for the conduction band. The dashed-black lines and solid-green lines both pertain to the low-energy model and are obtained by inserting slightly different sets of parameters into Eqs. (42) and (43).

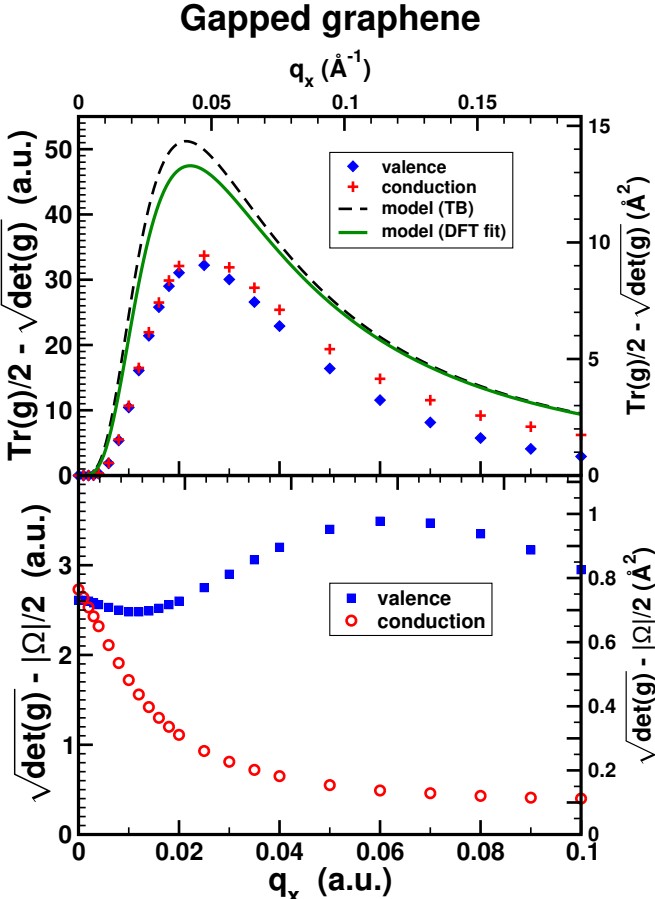

Figure 7: The relations between the trace and determinant of the quantum metric $g$ and the magnitude of the Berry curvature $\Omega$ are illustrated by plotting differences between those quantities as a function of the distance from K towards Γ. Notice the change of scale with respect to Fig. 5. In the top panel, the dashed-black lines and solid-green lines both pertain to the low-energy model and are obtained by inserting slightly different sets of parameters into Eq. (41). For the quantity plotted in the bottom panel, that model predicts a null value.

Next, we analyze the close connection between the real and imaginary parts of the quantum-geometric tensor, i.e., between the quantum metric and the Berry curvature. For nondegenerate states and two-dimensional materials, the quantum-geometric tensor becomes a $2 \times 2$ hermitian matrix. The three invariants of such a matrix are the determinant and trace of the real-symmetric part (quantum metric), and the value of the imaginary-antisymmetric part (Berry curvature). From the positive semi-definiteness of the quantum-geometric tensor, one obtains the following inequalities between the three invariants [52, 53],

$$\frac{1}{2}\operatorname{tr}(g) \geq \sqrt{\det g} \geq \frac{1}{2}|\Omega|. \tag{44}$$

(The first inequality is just between the arithmetic and geometric averages of the non-negative eigenvalues of the quantum metric.) Inspection of Eqs. 40 and 41 shows that the second inequality saturates for the low-energy model (this actually happens for any two-band model [53]), while the first only saturates in the limits $q \ll q_0$ and $q \gg q_0$. In Fig. 7, the quantities $\frac{1}{2}\operatorname{tr}(g) - \sqrt{\det g}$ and $\sqrt{\det g} - \frac{1}{2}|\Omega|$ are plotted as a function of wavevector distance from K towards Γ. Comparing the vertical scale of this figure with that of Fig. 5 shows that the inequalities are very close to being saturated, particularly in the case of $\sqrt{\det g} - \frac{1}{2}|\Omega|$ where the two-band model would predict a null value.

The two-band low-energy model fails to take into account the presence of other bands in the DFT calculation, which are responsible for the small nonzero values displayed in the bottom panel of Fig. 7. To estimate the corrections to the two-band model, note that the sum-over-states expression for $Q_{n\mathbf{k}}|u^{\alpha}_{dn\mathbf{k}}\rangle$ in Eq. 14 contains an energy-difference denominator. The Berry curvature and quantum metric will therefore depend on the inverse of the square of those energy differences. The smallest among them is the gap $\Delta = 0.28$ eV; the next relevant level (taking mirror symmetry into account) is about 9 eV away, and therefore $9/0.28 \sim 32$ times farther away. The corrections to the metric and Berry curvature should therefore be of the order of $(1/32)^2$, or about 0.1%. Indeed, this is the factor between the vertical scale of the bottom of Fig. 7 and that of Fig. 5. In the case of the inverse mass and orbital moment, the sum-over-states expressions depend on the inverse of the energy differences (not squared); as such, deviations from the two-band model are expected to be more pronounced, of the order of 3%. That is in fact almost the value by which the orbital magnetic moments at K in Fig. 6 deviate from their average. The difference between the absolute values of the conduction and valence effective masses is 1.2%.

# 5 Trigonal tellurium

Tellurium is one of the simplest crystals with interesting quantum-geometric band properties, which depend on a strong spin-orbit interaction in a reasonably heavy element, and on its chiral structure. As trigonal tellurium has six bands within less than 1 eV from the gap, it presents a more complex situation than the two bands of gapped graphene.

The self-consistent potential for the left-handed structure, with space group P$3_2$21, was calculated with the CPW2000 code using a modified Becke-Johnson meta-GGA functional [43]. The pseudopotential was relativistic Troullier-Martins [38] with a core radius of 2.6 atomic units, ground state configuration, and $s$, $p$, and $d$ channels. The local potential was a smoothed maximum of all channels, and the nonlocal part was converted to the KB form [26]. The crystal lattice constants were $a = 4.44$ Å and $c = 5.91$ Å, and the internal displacement parameter was $u = 0.269$ [54]. The self-consistent calculation used a kinetic energy cutoff of 18 Ha for the plane-wave expansion, and a $6 \times 6 \times 6$ uniform grid for BZ integration (63 **k** points once symmetry is taken into account). The Tran-Blaha parameter [43] was fixed at $c_{\text{TB}} = 1.10$. This value was chosen to give a band gap of 0.312 eV, the same as a previous calculation with the

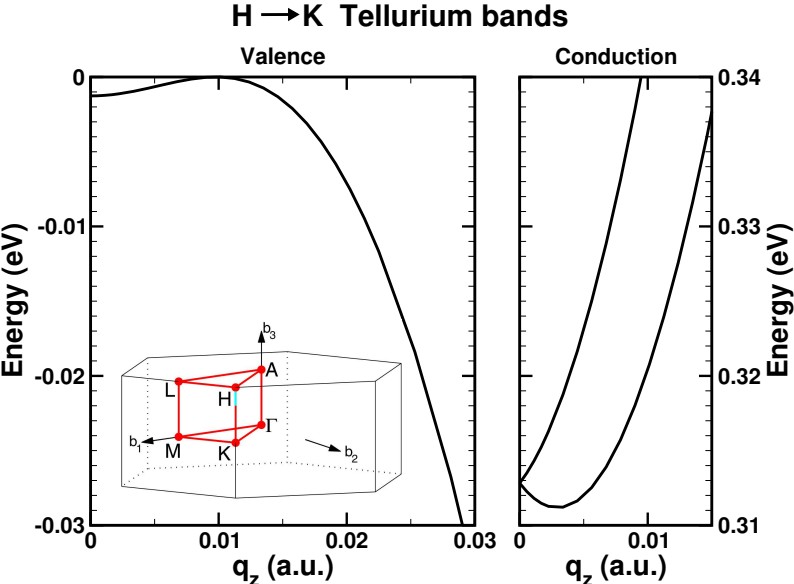

Figure 8: The energy bands of Te near the band gap are shown along the H–K line, near the H point ($q_z = 0$). The upper valence and lower conduction bands are shown in the left and right panels, respectively. On this highly detailed scale, the "camel back" structure of the valence band is clearly seen. The inset shows the hexagonal Brillouin zone. The short cyan segment on the H–K line highlights the considered region.

HSE06 hybrid functional [18], which is close to the 0.314 eV of a GW calculation [55], and to the experimental value of 0.323 eV [56].

The valence-band maximum (VBM) of trigonal Te, relevant for the naturally *p*-doped material, is located on the H–K line, the lateral edge of the hexagonal BZ, very close to the H points, which form the vertices of the hexagonal BZ: see inset of Fig. 8. The band dispersion displays a "camel back" feature which is clearly seen in the left panel of Fig. 8, with a local minimum at H (a saddle point in three dimensions), and a maximum about 0.02 Å$^{-1}$ close to it. In the present calculation, the depth of the minimum with respect to the maximum is 1.3 meV. The experimental value is 1.1 meV [57], the calculated value with an LAPW code and also a modified Becke-Johnson functional is 1.7 meV [16], and the calculated value with the VASP code and the HSE06 functional is 0.8 meV [18]. These are very small energies, and while the order of magnitude is consistent, there is some variation associated with different computational methods and functionals.

The Berry curvature near the VBM is shown in the top panel of Fig. 9 along the H–K line, as a function of the distance from the H point. Its sign depends on the handedness of the structure, and which of the two translationally-inequivalent H points is chosen. The solid line is the present calculation, and the squares are the results from a previous calculation with the HSE06 functional [18]. While the curves are similar, the values calculated with the present method are slightly smaller. We speculate the difference could be due to the use of different families of functionals: meta-GGA in the present case, hybrid functional in the previous calculation. The curvature vanishes at H by symmetry.

The quantum metric near the VBM is shown in the bottom panel of Fig. 9. At H the longitudinal component is significantly larger than the transverse component, but it decays much faster and becomes smaller outside the camel-back region.

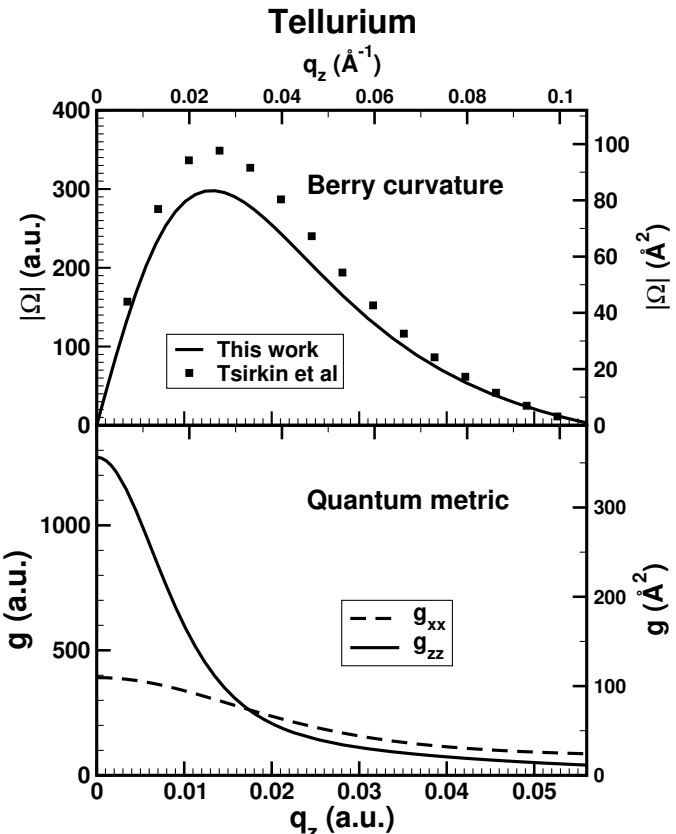

Figure 9: The Berry curvature (top panel) and quantum metric (bottom panel) of the upper valence band of Te are shown along the H–K line as a function of distance from the H point. In the top panel the solid line is the Berry curvature from the present calculation with a modified Becke-Johnson (mBJ) functional, and the squares are from a calculation by Tsirkin *et al.* [18] with the HSE06 hybrid functional. The bottom panel shows the quantum metric from the present calculation; the longitudinal component is represented by the solid line, and the transverse one by the dashed line. The left and bottom scales are in atomic units, while the top and right scales are in SI units.

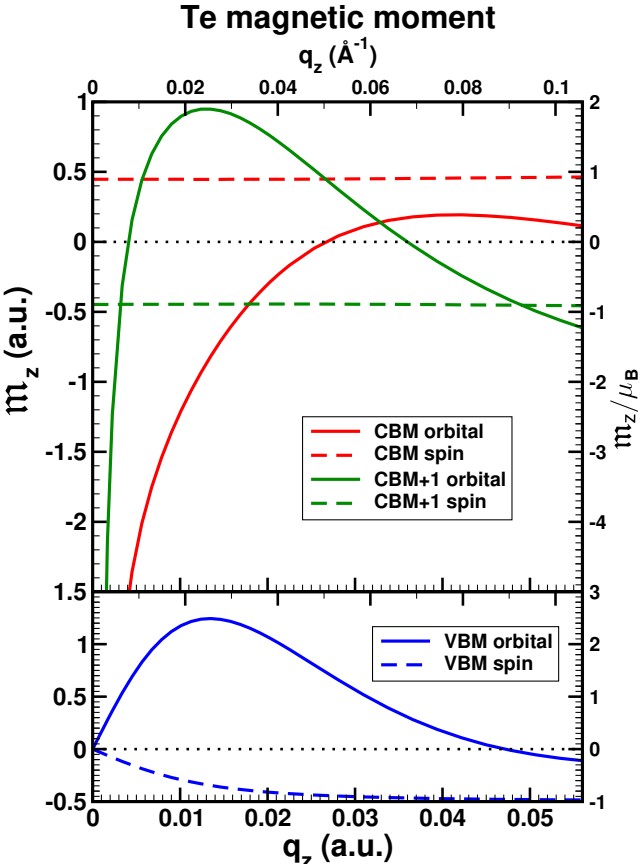

Figure 10: The orbital and spin magnetic moments of the upper valence band (VBM) and of two lowest conduction bands (CBM and CBM+1) of Te are shown along the H–K line as a function of distance from the H point. The solid and dashed lines are respectively the orbital and spin magnetic moments.

The magnetic moments near the VBM are shown in the bottom panel of Fig. 10. At the H point the upper valence band is nondegenerate, and both moments are zero by symmetry, just like the Berry curvature in Fig. 9. As the distance from H increases along the H–K line, the spin magnetic moment saturates to the Bohr magneton, while the orbital magnetic moment, which has the opposite sign, first increases and then decreases. In both cases the variations with wavevector are on the scale of the "camel back" feature in the band energies. The same quantities are shown in the top panel for the two bands near the conduction band minimum (CBM). Those bands are degenerate at H, where they have equal and opposite nearly saturated spin magnetizations. The variation of the spin magnetization with $k$ vector is negligible in the range of the figure. Instead, the orbital magnetic moment strongly diverges near H, as the two-band model from the previous section would predict in the limit of a vanishing gap. Near H the orbital moment of the lowest band has the opposite sign of the spin polarization, but away from H it decays and changes sign. In the second lowest conduction band, the orbital moment decays and changes sign much faster. For the upper valence band, the results are similar and consistent with a previous calculation [18].

The signs of $\Omega_z$ and $\mathfrak{m}_z$ change depending on the handedness of the crystal, on which inequivalent H point one considers, and, in Fig. 9, on whether $q_z$ is on the $+z$ or $-z$ direction. The relative signs of the orbital and spin magnetic moments and Berry curvature between the different bands do not change, there is only a global sign flip. As mentioned previously, the plotted results are for the left-handed structure. With the usual $2\pi/3$ angle between the in-plane primitive lattice vectors, and with the first Te atom at ($u = 0.269, 0, 0$) in lattice coordinates, the H point of the figure is at $(1/3, 1/3, 1/2)$ in reciprocal lattice coordinates, at a 60° angle in the basal plane with respect to the projection of the atomic positions. The $q_z$ is in the negative direction as indicated in the inset of Fig. 8, $\mathbf{k} = \mathbf{b}_1/3 + \mathbf{b}_2/3 + (1/2 - q_z)\mathbf{b}_3$, and $\Omega_z$ along that line has negative values.

The components of the inverse effective mass of the upper valence band of Te are shown in Fig. 11 along the H–K line. The longitudinal mass at H is electron-like while the transverse is hole-like, as expected from a saddle point.

Along that line, the longitudinal effective mass changes sign going over the "hump" of the "camel back", and at the top of the valence band it is hole-like. The maximum is 0.0185 Å$^{-1}$ away from H, where the longitudinal mass is 0.311 $m_e$ and the transverse mass is 0.129 $m_e$. The corresponding values from the LAPW mBJ calculation are 0.251 $m_e$ and 0.098 $m_e$ [16], while the experimental values are 0.220 $m_e$ and 0.108 $m_e$ [58].

# 6   Conclusions

The numerically precise calculation of the basic quantum-geometric properties of crystals, namely Berry curvature, quantum metric, orbital magnetic moment and effective mass, was successfully implemented in a first-principles pseudopotential plane-wave code. The adopted procedure is as follows. For a given effective potential and wavevector $\mathbf{k}$, the first derivatives of the cell-periodic Bloch wavefunctions $u_{dn}(\mathbf{k})$ with respect to wavevector $\mathbf{k}$ are obtained from perturbation theory. This is achieved by solving iteratively a Sternheimer equation, taking into account energy-level degeneracies and using a stable algorithm. Once those wavefunction derivatives are obtained, the four quantum-geometric quantities are readily determined as the real and imaginary parts of two complex objects: the quantum-geometric tensor $T_{dd'}^{\alpha\beta}$ of Eq. (22) (quantum metric and the Berry curvature), and the closely-related mass-moment tensor $\Gamma_{dd'}^{\alpha\beta}$ of Eq. (28) (inverse effective mass and orbital moment).

Effective masses can also be calculated by finite differences. The excellent agreement between perturbation theory and finite-differences values for the effective masses of Si and GaAs

**Tellurium inverse effective mass**

Figure 11: The inverse effective mass of the top of the valence band of Te is shown along the H–K line as a function of distance from the H point. The solid line is the longitudinal component, and the dashed line is the transverse component. Consistent with the "camel-back" feature of that band, the longitudinal mass changes from positive, electron-like, to negative, hole-like. The transverse mass remains hole-like in almost the entire range of the plot.

shows that they are precisely calculated with our implementation. This in turn implies a precise determination of the underlying wavefunction derivatives, and therefore, the numerical precision of the other quantum-geometric quantities as well.

In both Si and GaAs, the top of the valence band has nontrivial degeneracies. As a result, the $T$ and $\Gamma$ tensors have a total of four indices: two for the spatial dimensions ($\alpha\beta$), and two for the level degeneracy ($dd'$). The calculation of the direction-dependent transport-equivalent effective masses illustrates how to extract physical properties from those complex objects, taking into account the subtleties of perturbation theory for degenerate states.

Calculations of quantum-geometric quantities for gapped graphene were performed by adding a bespoke artificial potential to first-principles graphene. As the two bands near the opened gap are reasonably decoupled from other bands, in particular once mirror-symmetry selection rules are accounted for, the first-principles results can be compared to a well-known two-band low-energy model. With appropriate parameters, that model closely reproduced the calculated DFT values, apart from some deviations that were discussed. Such detailed comparison between low-energy models and direct *ab initio* calculations has not, to our knowledge, been reported in the literature.

Trigonal tellurium, one of the simplest crystals with interesting quantum geometric properties, was chosen as the final example. The properties were calculated in the neighborhood of the valence and conduction band edges. For the "camel back" region of the valence-band maximum, the calculated orbital magnetic moment was similar to the results of a previous calculation. In the low-lying conduction bands, the orbital moment near the H point was found to have the opposite sign from that of the upper valence band. This could have interesting implications for nonlinear magneto-transport properties such as electrical magnetochiral anisotropy [59].

In view of current interest in the quantum geometry of electron states in crystals and its physical manifestations, we believe that the formalism and implementation presented in this work could find multiple applications.

# Acknowledgments

Work by J. L. M. and C. L. R. was supported by grant EXPL-FIS-MAC-1334-2021 from the Portuguese Science and Technology Foundation (FCT). They wish to acknowledge FCT for funding the Research Unit INESC MN (UID/05367/2020) through Plurianual, Base and Programatico financing. Work by I. S. was supported by Grant No. PID2021-129035NB-I00 funded by MCIN/AEI/10.13039/501100011033 and by ERDF/EU. The authors thank C. J. Augusto of Quantum Semiconductor LLC for useful discussions throughout this work, and for a careful reading of the manuscript.

# References

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
