# Peer review of "Precise quantum-geometric electronic properties from first principles"

_SciPost Physics_

## Round 1 · Referee Report · Anonymous (Referee 1) · 2025-8-8

Strengths

1) new and original methods to calculate quantum geometry quantities 2) comparison with existing methods and models 3) a code with examples is available

Weaknesses

1) a simple scheme/flowchart of their method could make it more readable

Report

In this manuscript, the authors present an original approach for calculating quantum geometric properties from first principles, combining perturbation theory with the Sternheimer equation. The different steps involved in obtaining quantum-geometric properties from the Sternheimer equation are clearly explained. The authors tested their approach on different systems and compared their results with those obtained using finite differences, analytic models, K-dot-P and perturbation theory. I found their results to be correct and their new approach interesting. For this reason, this manuscript could be published in its current form. However, the authors could address the following minor questions/remarks in a revised version of their manuscript:

1) The authors could draw a simple scheme or flowchart of their method to make it more accessible to readers.

2) What is the value of k₀ in the gallium arsenide example?

3) May the authors comment on the scaling of their Sternheimer equation with system size, compared with perturbation theory or finite differences?

4) In the bottom panel of Fig. 7, the authors discuss a quantity that is different from zero in DFT due to the presence of additional bands compared to the simple 2 bands model. Could the presence of these bands also be the origin of the small differences in Figs 5 and 6? Could the authors comment on this point?

Requested changes

see report

Recommendation

Ask for minor revision

  • validity: high
  • significance: high
  • originality: high
  • clarity: top
  • formatting: excellent
  • grammar: excellent

Author:  José Luís Martins  on 2025-09-03  [id 5772]

(in reply to Report 1 on 2025-08-08)

1) We followed the suggestion, and included such a flowchart as the new Fig. 1. (page 8)

2) The value, 10^{-4} inverse Angstroms, is now specified in the text (p. 14 line 344).

3) We added a new subsection, Sec. 3.3, comparing perturbation theory and finite differences. It includes the comment that an iterative solution of Sternheimer equation scales with system size in the same way as an iterative diagonalization of the Hamiltonian. (p. 16, lines 366-384)

We did not elaborate on the point of view of the user. In general, once one has access to a perturbation-theory implementation, one does not want to deal with finite differences again.

4) We agree with the referee, and added a sentence (p. 22 lines 500-502) saying that those other bands are also responsible for the small differences seen in Figs. 6 and 7 of the revised manuscript (Figs. 5 and 6 of the original manuscript).

---

## Round 1 · Referee Report · Anonymous (Referee 2) · 2025-8-10

Strengths

1- New numerical implementation of quantum-geometric properties of crystals, i.e., Berry curvature, quantum metric, orbital magnetic moment and effective mass, in a pseudopotential planewave DFT calculation (CPW2000 code)

Report

This paper propose a numerical implementation of quantum-geometric properties of crystals, i.e., Berry curvature, quantum metric, orbital magnetic moment and effective mass, in a pseudopotential planewave DFT calculation (CPW2000 code), using perturbation theory. In the current context of condensed matter physics, which is increasingly studying quantities that enable us to qualify the topological properties of materials, this original and innovative work seems very interesting and promising.

The manuscript is well presented, with a first section that explains the method used in a fairly educational way. The second section presents three applications of their method and a detailed comparison of the results with other methods. The authors successfully tested their method on the effective masses of silicon (Si) and gallium arsenide (GaAs) semiconductors, as well as on the quantum-geometric quantities of gapped graphene and trigonal tellurium. This seems to me to be a significant advance in the field of DFT numerical calculations, which should generate great interest in future studies of the electronic properties of materials. So I think it deserves to be published in sciPost Physics.

Requested changes

I only have one question and a few minor comments that the authors may wish to consider in order to improve the understanding of their manuscript.

1- In both the abstract and the introduction, the authors emphasise that they paid particular attention to cases involving degenerate states. Indeed, the numerical calculations of quantum-geometric quantities are often challenging in the presence of degenerate states. Perhaps I have misunderstood something, but I do not see how the method used is more effective than alternative methods. Some clarification on this point would be welcome.

Minor comments/typos: 2 - Fig. 2: The meaning of the dashed magenta lines is not explained in the legend.

3 - Ligne 387: “(…) a 6 × 6 × 2 uniform grid for BZ integration.” Why use 2 in the direction perpendicular to graphene when it is a 2-dimensional system?

4 - Ref. [8]: the link “2501.00098” is broken and useless.

5 - Refs. [26]: the correct DOI is: 10.1103/PhysRevLett.48.1425.

6 - Refs. [28]: the correct DOI is: 10.1103/PhysRevLett.55.2471.

Recommendation

Publish (easily meets expectations and criteria for this Journal; among top 50%)

  • validity: high
  • significance: high
  • originality: high
  • clarity: high
  • formatting: good
  • grammar: -

Author:  José Luís Martins  on 2025-09-03  [id 5771]

(in reply to Report 2 on 2025-08-10)

1) The referee is partially correct. Re-reading the original abstract, it seems to suggest that the difficulty is attached to the solution of the Sternheimer equation. Instead, our intention was to convey the idea that it was a difficulty of the entire implementation.

We removed that sentence from the abstract. In the revised Introduction, we made it clear that the difficulty is in the whole implementation, and why it is more difficult. (p. 3 lines 53-57)

2) Added a sentence in the figure caption (Fig. 3 p. 13 in the revised manuscript) clarifying the meaning of that line.

3) Added one phrase in the text (p. 16-17 lines 411-413). The physics behind the choice both subtle and simple. A 1D tight binding band has the energy E(k) = E_a - t cos(k a), where t -> 0 as the slab distance is increased. Choosing k = (1/4) (2 pi/a), which is the "x 2" with 0.5 shift, annuls the cos() term. We double checked that the 6x6x1 does indeed converge slower, although for the layer separation used in the calculation, the differences are minor.

4) It is a problem with the processing of the eprint field in bibtex. The normal URL works.

5) We have implemented this correction. (New ref. 27, line 705)

6) We have implemented this correction. (New ref. 29, line 709)

---

## Editorial Decision

resubmitted